# Enhanced voltage generation through electrolyte flow on liquid-filled surfaces

B. Fan[1], A. Bhattacharya[1] & P.R. Bandaru[1,2]

The generation of electrical voltage through the flow of an electrolyte over a charged surface may be used for energy transduction. Here, we show that enhanced electrical potential differences (i.e., streaming potential) may be obtained through the flow of salt water on liquid-filled surfaces that are infiltrated with a lower dielectric constant liquid, such as oil, to harness electrolyte slip and associated surface charge. A record-high figure of merit, in terms of the voltage generated per unit applied pressure, of 0.043 mV Pa$^{-1}$ is obtained through the use of the liquid-filled surfaces. In comparison with air-filled surfaces, the figure of merit associated with the liquid-filled surface increases by a factor of 1.4. These results lay the basis for innovative surface charge engineering methodology for the study of electrokinetic phenomena at the microscale, with possible application in new electrical power sources.

[1] Department of Mechanical Engineering, University of California, San Diego, La Jolla 92093-0411 CA, USA. [2] Program in Materials Science, University of California, San Diego, La Jolla 92093-0411 CA, USA. Correspondence and requests for materials should be addressed to P.R.B. (email: pbandaru@ucsd.edu)

The motion of an aqueous electrolyte, such as salt water, over a surface may be harnessed for the generation of electrical voltage[1]. Indeed, the exploration of such electrokinetic phenomena and their possible use for energy conversion have a long history, extending over the past two centuries (from Morrison & Osterle[2] and cited references). The relevant potential difference, termed as streaming potential ($V_s$), arises due to the relative motion of charged[3] species in the electrolyte with respect to the fluid channel substrate with residual charges[4]. While superhydrophobic (SH) surfaces[5] have been posited to increase the ion velocity and the resultant potential difference, such enhancement has not been observed to date due to the inability of the air in the SH surfaces to hold charge[6]. Here, we monitor the flow and measure the $V_s$ of a NaCl-based electrolyte (consisting of Na$^+$ and Cl$^-$ dissolved in water) with a higher dielectric constant ($\varepsilon$) over a lower $\varepsilon$ solid substrate/surface with an induced negative charge[7,8] in a microfluidic channel[9]. The electrically compensating positive counterions reside within the adjacent electrical double layer (EDL)[10], which consists of an inner layer with fixed counterions adsorbed onto the surface and a diffuse layer with mobile counterions, with a thickness of the order of the Debye length ($\lambda_D$). On application of a pressure difference ($\Delta P$) across the two ends of the channel, say, through a mechanical pump, Fig. 1a–c, the electrolyte flow would be mainly constituted by the mobile counterions in the diffuse layer with the consequent charge separation yielding a streaming potential ($V_s$) that is proportional to the $\varepsilon$ (= $\varepsilon_o\varepsilon_r$- with $\varepsilon_o = 8.854 \cdot 10^{-12}$ C$^2$ N$^{-1}$ m$^{-2}$ as the free-space permittivity, and $\varepsilon_r$ as the relative permittivity of electrolyte e.g., ~ 80 for 0.1 mM L$^{-1}$ NaCl solution) and the zeta potential ($\zeta$) at the edge of the shear plane where the mobile ion motion occurs and varies inversely with the dynamic viscosity of the electrolyte ($\eta$) and bulk electrolyte conductivity ($\kappa$), as described through the Helmholtz–Smoluchowski model[7,11]:

$$V_s = \frac{\epsilon\zeta}{\eta\kappa}\Delta P \qquad (1)$$

The $\zeta$ is considered close to the substrate–electrolyte liquid interface and is significant, in that it determines the electrical potential that may be utilized[7,8] for the $V_s$. In the flowing electrolyte itself, there is a close to exponential decay of the surface potential over[10] $\lambda_D$. It should be noted that Eq. (1) is based[12] on the assumptions of negligible surface (/substrate) conductivity and a very small EDL thickness[13], with Poiseuille flow of the electrolyte. Moreover, there is an implicit assumption of the no-slip boundary condition[14], where the liquid adjacent to the substrate wall has a zero velocity, with a finite flow velocity only at a certain distance (corresponding to the shear plane) into the fluid. Consequently, $\zeta$ would be related to the electrical potential at the edge of the shear plane in electrokinetic flows over smooth surfaces[7,12]. Most work on harnessing the $V_s$ through electrokinetic effects, to date, has been concerned with fluid flow over smooth surfaces (where the scale of roughness is smaller than $\lambda_D$), and consequently very small potential differences of the order of 18 mV may be predicted and obtained[11], in correspondence to Eq. (1), i.e., for 0.1 mM L$^{-1}$ NaCl, with $\varepsilon_r$ ~ 80, $\eta$ ~ $10^{-3}$ Pa·s, $\zeta$ ~ 25 mV, $\kappa$ ~ $10^{-3}$ S m$^{-1}$, with $\Delta P$ ~ 1000 Pa.

With the objective of obtaining significantly larger $V_s$, we indicate briefly the principles of our approach, which first involved the modulation of the effective $\zeta$ through introducing fluid slip via groove-patterned surfaces (in air-/liquid-filled surfaces) and modifying surface charges at the substrate surface. It has previously been considered, based on molecular dynamics simulations[15], that slip may mobilize the Stern layer, significantly enhancing the $\zeta$. The broad concept is that the shear plane/surface of shear (at which the electrolyte flow velocity is zero) is to be

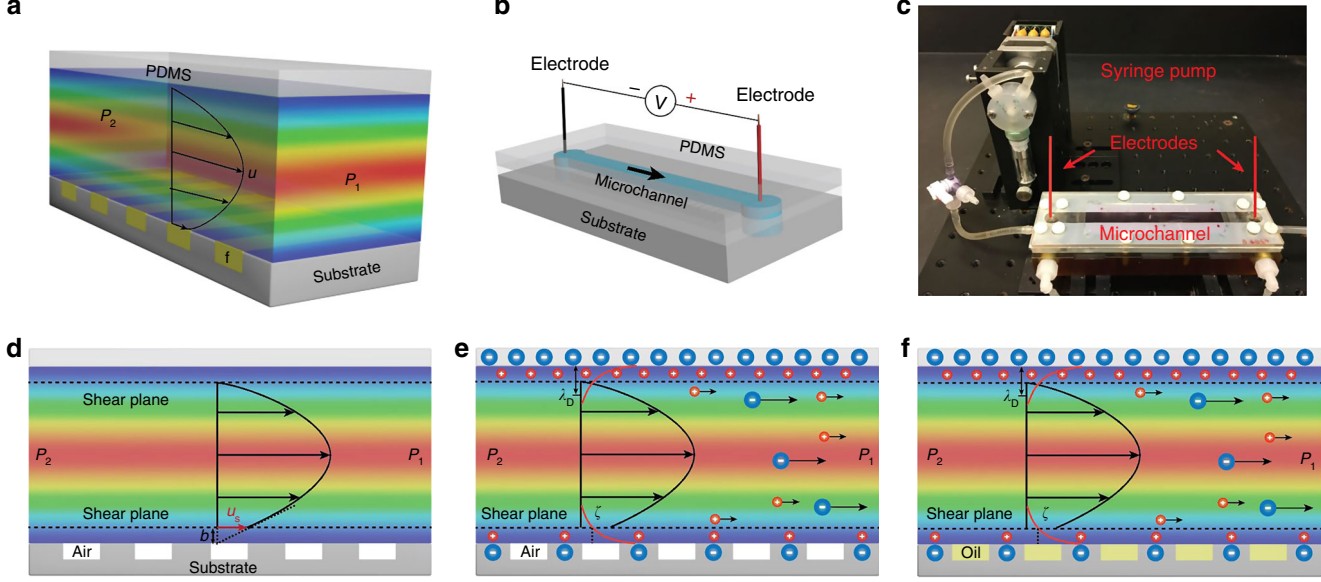

**Fig. 1** Electrokinetics over patterned and fluid-filled surfaces. The related phenomena were investigated **a** through monitoring the streaming potential ($V_s$) of salt water, under pressure driven flow, in a microchannel (of length 11.8 cm, width of 0.9 cm and height of ~250 μm). The upper surface of the channel was poly-dimethyl siloxane (PDMS), while the lower fluid–solid surface was engineered with the fluid (f), such as oil or air. The velocity ($u$) profile with fluid slip due to the pressure difference ($P_2$–$P_1$) applied by syringe pump is indicated. **b** The schematic of the flow arrangement in the microchannel and its **c** experimental realization, wherein the top and the bottom surfaces were separated by a silicone rubber spacer to adjust the microchannel height, and Ag/AgCl electrodes were inserted in reservoirs at either end to measure the $V_s$. While **d** liquid flow over an air-filled surface (AFS) yields a finite slip velocity ($u_s$) and slip length ($b$), the **e** flow of an electrolyte such as salt water yields a finite $V_s$ due to surface charge (indicated as negative, here)-induced influence on the electrolyte ions. **f** The $V_s$ can be made significantly larger when the air is replaced by a water immiscible liquid, such as oil yielding a liquid-filled surface (LFS)

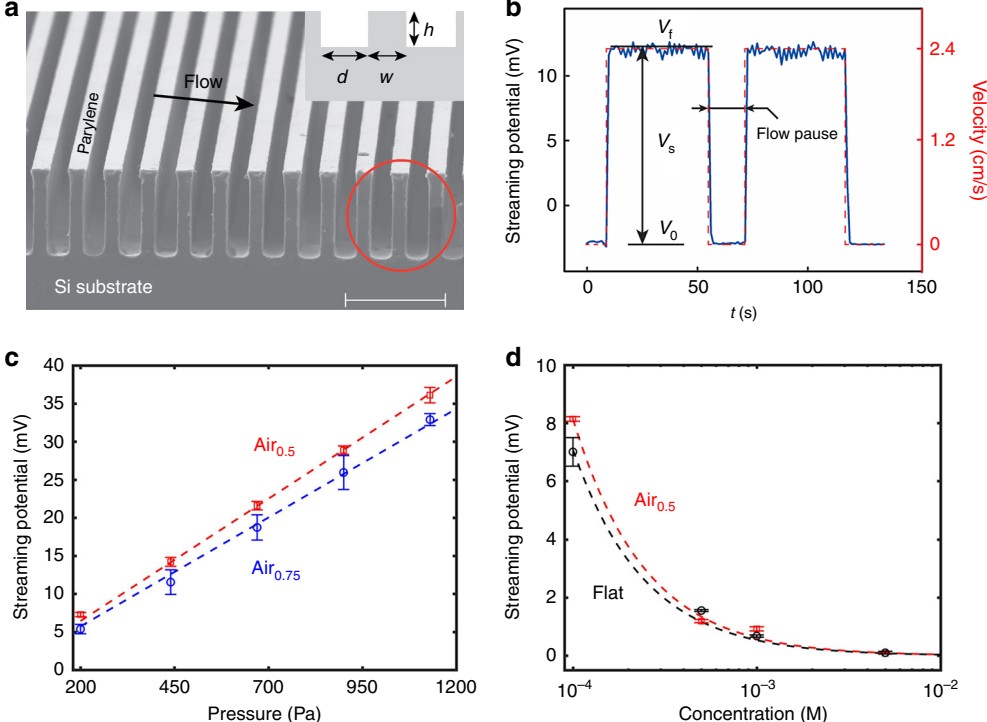

**Fig. 2** Streaming potential of air-filled surfaces. **a** A scanning electron microscopy (SEM) micrograph of the groove-patterned surfaces with silicon as the substrate covered by parylene-C (thickness $\approx 600$ nm) parameterized on the basis of the solid surface width ($w = 18$ μm), groove width ($d = 18$ μm), pattern height ($h \approx 95$ μm), and the air fraction: $\phi_{air}$ [ $= d/(d + w) = 0.5$] and a solid fraction: $\phi_s = 1 - \phi_{air}$. The flow direction is perpendicular to the grooves and indicated by the arrow. Scale bar is 100 μm. **b** The typical response of streaming potential ($V_s$) was measured, for a $\phi_s = 0.5$ air-filled surface (AFS), through a syringe pump-driven pressure, e.g., with an applied pressure of 440 Pa, 0.1 mM L$^{-1}$ NaCl here. The red dotted line indicated the average flow velocity in the channel. The measured voltage difference between a baseline value ($V_o$) and a final value ($V_f$) was considered as $V_s$. **c** The measured $V_s$ (0.1 mM NaCl solution) on an AFS (i.e., Air$_{0.5}$, the subscript indicating the $\phi_{air}$) scales linearly with the applied pressure (in the range of 200–1200 Pa), in accord with the Helmholtz–Smoluchowski model (Eq. 1) with a lower value for $V_s$ obtained for a smaller (/larger) $\phi_s$ (/$\phi_{air}$) and rationalized as due to the absence of surface charges and contributing zeta potential $\zeta$, in the air regions. **d** The $V_s$ scales with the salt water electrolyte conductivity and related concentration ($l_c$) through a log [$l_c$]/ [$l_c$] variation for both AFS (Air$_{0.5}$) and parylene-coated flat surface (Flat) (concentration varying from 0.1 mM L$^{-1}$ to 5 mM L$^{-1}$ at 200 Pa for **d**)

moved as close as possible to the substrate. As the surface electrical potential decays away from the substrate, the closer the shear plane is to the substrate boundary, the larger the $\zeta$ with concomitantly increased $V_s$. Here, the zero velocity boundary condition at the surface ($y = 0$) would be replaced with a Navier slip condition, $u_s (y = 0) = b \frac{\partial u(y=0)}{\partial y}$ with $u_s$ as the slip velocity and $b$ as the slip length[16,17], Fig. 1d. The larger the $b$, the greater the fluid velocity at the surface by a factor of $\left(1 + \frac{b}{\lambda_D}\right)$. A related apparent increase[18] in the $\zeta$ and the $V_s$ may then be consequently obtained[19]. As much relatively large values of $b$ of the order of 10 μm were thought to be obtained through the use of SH surfaces[20], and given a $\lambda_D$ of 9.6 nm with 1 mM NaCl, a $b/\lambda_D$ of ~1000 is estimated, yielding large $V_s$. While non-uniform surface conduction[21], e.g., as related to slip to no-slip transitions, may reduce such an enhancement, a finite $b$ would yet contribute to $V_s$ and may be aided by surface charge induced forces.

To further understand the issues related to electrokinetics over surfaces with slip, we conducted experiments (see Figs. 1b, c) where the electrolyte flow was driven in a microfluidic channel (of length: 11.8 cm, width: 0.9 cm, and height: 250 μm) by a syringe pump, with $\Delta P$ varied in the range of 0–1200 Pa (see Methods) to measure the $V_s$ over ridged pattern surface morphology, termed as AFS (air-filled surface), Fig. 1e. The substrate material of AFS is silicon, coated by 600-nm hydrophobic parylene-C. The

patterns were characterized through an interstitial/air fraction: $\phi_{air}$, through the ratio [$= d/(d + w)$], with an average ridge width ($w$) and spacing ($d$), Fig. 2a. We report results on samples with $\phi_{air} = 0.5$, $d = w = 18$ μm for $\phi_{air} = 0.75$, $d = 27$ μm, and $w = 9$ μm (see Methods for channel design and fabrication-related considerations, as well as the optimization and choice of the parameters in Supplementary Note 1).

We show that a record large figure of merit, in terms of the voltage generated per unit applied pressure, is obtained by using a liquid to fill the grooves in comparison to air-filled grooves.

## Results

**Voltage generation on air-filled surfaces.** Considering the flow of electrolyte over air-filled surfaces, the consequent voltage–time trace, Fig. 2b, was used to determine the $V_s$, which was monitored through a high-resistance (> 10 GΩ) voltmeter (Keithley 2700) connected across the Ag/AgCl electrodes placed at either ends of the channel, Fig. 1c. However, a significant enhancement was not obtained for AFS, with an estimated[5] $b$ of ~10 μm. It should be noted that a lower $V_s$ was obtained for a larger $\phi_{air}$ (i.e., smaller $\phi_s$), for which a larger $b$ would be expected, Fig. 2c. Moreover, very little increase over a flat unpatterned surface was obtained comparing Air$_{0.5}$ and Flat, Fig. 2d. The lower $V_s$ of AFS may be rationalized as due to the absence of a surface charge in the air regions[13], which overwhelms the contribution of the slip.

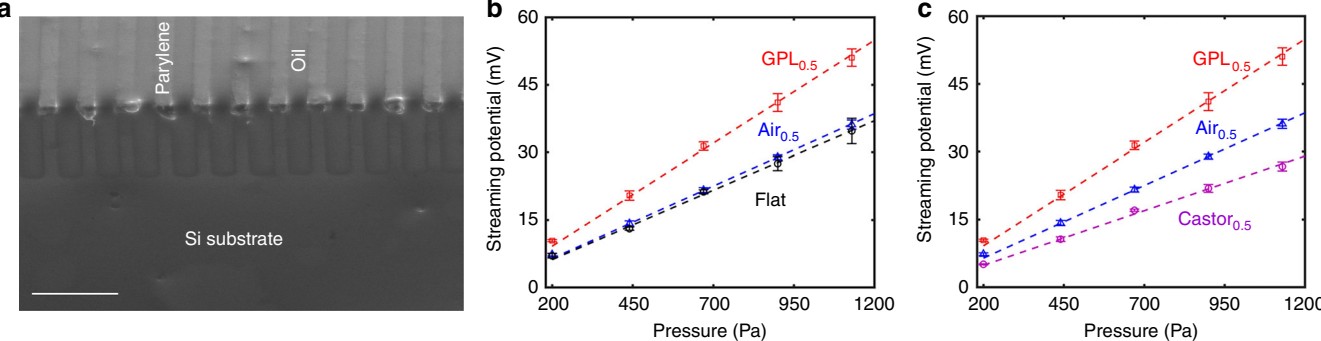

**Fig. 3** Enhanced streaming potential at liquid-filled surfaces. Enhanced streaming potential ($V_s$) was obtained by **a** filling the grooves with salt water immiscible oil, e.g., Krytox GPL 104, (scale bar is 100 μm); **b** a significantly larger $V_s$ was obtained for an GPL 104 oil-filled surface (i.e., GPL$_{0.5}$, with the subscript now indicating the interstitial fraction of oil), compared to one with air (Air$_{0.5}$) and even compared to a Flat solid substrate (all measured using 0.1 mM L$^{-1}$ NaCl solution with applied pressure as 1200 Pa). **c** The $V_s$ could be larger or smaller compared with the control air-filled surfaces (AFS), depending on the choice of the liquid in the LFS, e.g., GPL oil and castor oil (Castor$_{0.5}$), respectively

While the data here seems to indicate the general validity of Eq. (1) in terms of a linear $V_s$-$\Delta P$ plot, the slope of the obtained experimental curves would then be directly connected to $\varepsilon$, $\zeta$, $\eta$, and $\kappa$. Such parameters are then intimately connected, i.e., while independent variation would be difficult to discern experimentally, theoretical or computational input (which is outside the purview of the present experimental work) may be able to yield guidance for explicit dependence. The net $V_s$ has been construed to be related to a weighted summation of $\zeta$ in the no-slip region (over the solid fraction of the surface, i.e., $\zeta_{NS}$) and the slip region (i.e., over the air of the AFS, i.e., $\zeta_S$)[6]. The very small $\zeta_S$ would adversely affect the $V_s$ and as seen through our experiments overwhelms the slip length-related contribution. Related correspondence of the obtained $V_s$ to $\zeta$ as well as the electrolyte conductivity, which is proportional to $I_c$ (the counter-ion concentration), was determined through a log $[I_c]/[I_c]$ dependence, Fig. 2d.

**Increased voltage on liquid-filled surfaces**. Subsequently, we deployed patterned surfaces where the air pockets were infiltrated by a water-immiscible liquid, e.g., oil, of a higher density compared with the NaCl electrolyte, Fig. 3a. Such liquid-filled surfaces (LFS) were hypothesized to yield a finite $\zeta_S$ at the oil interfaces, in addition to a non-zero $b$. The LFS were fabricated through replacing the air in groove-patterned arrays by water-immiscible liquid, such as oil, and may be designed to exhibit various degrees of fluid slip[22–24], as well as electrical voltages that are significantly larger than those on air-filled or SH surfaces. Oil filling was found to be reliable and stable (see Methods).

Additionally, the use of oil with a lower $\varepsilon_r$ compared with the flowing electrolyte would mean that the electric field from the wall is effectively propagated. The details related to the (i) choice of liquid for the LFS along with the estimation of the (ii) $\zeta$, and the (iii) $b$ are given in Methods, Supplementary Note 2 and Supplementary Note 3, respectively. Using Krytox GPL 104 oil ($\varepsilon_r$ ~ 2.1, $\eta$ ~ 340 cP), a significant enhancement of the $V_s$ by 50% to around ~52 mV, compared with that of Air$_{0.5}$, was observed, Fig. 3b, attesting to the utility of the LFS. However, it was noted that the obtained $V_s$ critically depends on the choice of the oil in the LFS. Broadly, a larger $\varepsilon_{oil}$ seems to yield a lower $V_s$. For instance, a factor of two smaller value was obtained for a castor oil ($\varepsilon_r$ ~ 4.7, $\eta$ ~ 312 cP)-filled LFS, Fig. 3c. The enhanced Van der Waals energy of interaction[25] between the oil and electrolyte, proportional to the dielectric constants, and the related mutual electric polarizabilities causes a larger friction. The consequent

movement of the shear plane away from the solid substrate would result in a lower zeta potential and $V_s$.

The electrolyte flow over the LFS, as well as the adjacent solid surface were parameterized[26] through the use of a material-dependent surface charge density ($\sigma$) related to $\zeta = \frac{\lambda_D \sigma}{\epsilon}$ and fluid slip at the respective interfaces. From literature, for the top surface of the channel (constituted from poly-dimethyl siloxane, PDMS), $\sigma_{PDMS}$[27] ~18 mC m$^{-2}$, while for the bottom-patterned surfaces, we use $\sigma_{oil}$[28] ~1.8 mC m$^{-2}$ and $\sigma_{parylene}$[29] ~3.6 mC m$^{-2}$. The $\sigma$ and the velocity slip is inhomogeneous along the length of the channel, but constant along the channel width. The no-slip hydrodynamic boundary condition was assumed to hold true at all solid–electrolyte interfaces, while a Navier slip boundary condition over the flowing electrolyte–oil interface was assumed, with a finite slip velocity: $u_s$ and slip length: $b$. For a hydrophobic surface, the $b$ may be approximated[5] to be of the order of $d$, i.e., ~ 18 μm. A correspondence to a Cassie state-like channel flow (with no electrolyte penetration into the grooves of the patterned surface) for LFS surface may be seen. It should be noted that simulations considering the continuity of tangential shear stress across the electrolyte–oil interface also obtained the $V_s$ very close (within 1%) to the results obtained through an assumption of a slip length.

**Modeling of electrolyte flow over the liquid-filled surfaces**. We partition[12] the pressure gradient driven electrolyte flow, following experiment, over the LFS as the superposition of (i) hydrodynamic flow, with fluid slip over electrically neutral surfaces, and (ii) an electrokinetic flow, with no-slip fluid flow over flat electrically charged surfaces, Fig. 4a–c. A multiphysics model coupling the Nernst–Planck–Poisson (N-P-P) equation with the Stokes Equation was deployed to determine the volumetric charge density profiles (the details of the governing equations, assumptions, and the boundary conditions are given in Supplementary Note 4 and Supplementary Note 5). The velocity profile combined with the simulated volumetric charge density profile along the channel, as indicated in Fig. 4d, arises due to the difference of ionic concentrations between the counterions and co-ions. The resultant horizontal electric field is responsible for the observed $V_s$. The calculated average electrolyte velocity of 6 cm s$^{-1}$ is in agreement with that expected from Poiseuille flow along the channel, with the estimated $u_s$ at the oil–electrolyte interface of ~2 cm s$^{-1}$ at $\Delta P$ ~1200 Pa. Additionally and given the relatively large groove widths, the influence of the surface conductivity on the obtained results would be relatively small[19].

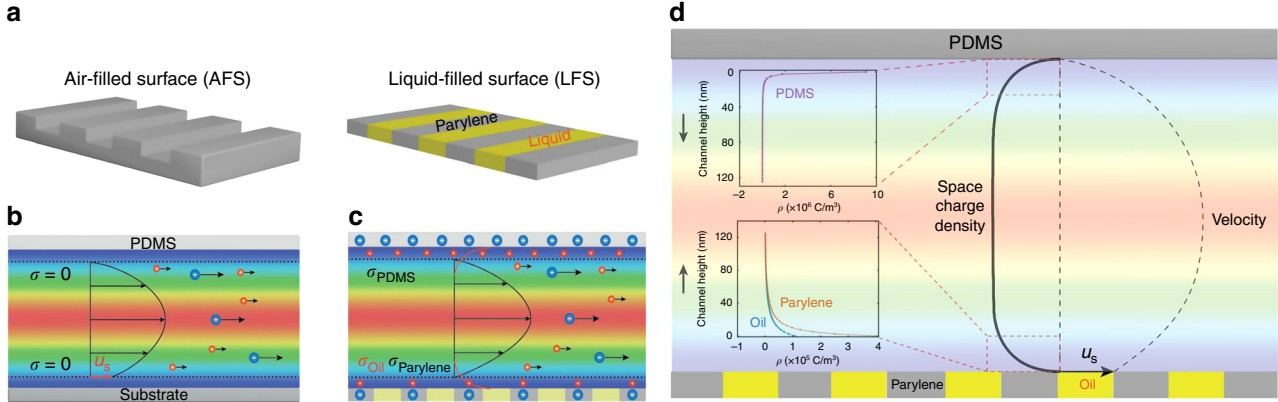

**Fig. 4** Parametrization and modeling of charged fluid–solid surfaces. **a** The surface charge density ($\sigma$) of an air-filled surface (AFS) varies between zero (at the air) and a finite value at the solid, while a nominally planar liquid-filled surface (LFS) maintains a larger value of the $\sigma$. The electrokinetics on the LFS was considered in terms of a superposition of **b** hydrodynamic flow, with fluid slip over electrically neutral surfaces, and **c** an electrokinetic flow, with no-slip fluid flow over electrically charged surfaces, and solving the coupled Nernst–Planck–Poisson (NPP) equation along with the Stokes Equation to determine the **d** velocity profile, as well as the volumetric charge density ($\rho$) profiles. The insets indicate the computed variation of the $\rho$ at the top poly-dimethyl siloxane (PDMS) surface and the bottom LFS

## Discussion

The values of the streaming potential arising from the simulations are in excellent agreement with those observed experimentally through considering a reduced $\varepsilon_r \sim 28$ over a distance $\sim 3\lambda_D$ proximate to the LFS and an $\varepsilon_r \sim 78$ and beyond, correspondent to the bulk solution. Such aspect is in accord with the use of different values of the dielectric constant for the inner and outer regions of a solid–solution interface[12]. The LFS, consisting of both oil as well as solid surfaces, may yield an uneven shear plane with fascinating implications for novel electrokinetic phenomena[30,31], such as localized concentration polarization[19], etc. Our work also indicates that a heterogeneous surface could be patterned through the use of discrete surface charge density or applied potentials, mimicking a SH surface and making a connection with electro-wetting applications, as well as with significant literature on the modeling and parameterization of liquid flows on striped surfaces that exhibit alternate regions of slip and no-slip, with[32,33] and without[34] surface charge.

A more detailed investigation of several issues related to the interplay of enhanced surface charge as well as increased slip velocity at the flowing electrolyte–LFS interface would then be warranted. It has been indicated, for instance, that an increased $V_s$ would result due to the larger ion convection currents[35]. Additionally, viscous effects may be playing a major role[36,37]. As the coefficient in Eq. 1 intimately connects the $\varepsilon_r$, and the $\eta$, a decrease in the former may not be readily deconvoluted from an increase in the latter experimentally. In our numerical simulations, an increase in the $\eta$ by a factor of $\sim 2.5$ over the equivalent length scale (from a bulk value of 0.89 cP to 2.19 cP) was necessary to obtain the experimentally observed streaming potentials, if the $\varepsilon_r$ is $\sim 78$.

Our experiments have deep scientific implications underlying fluid flow interfacing with both air and liquid, as well as fluid shear at a surface, and electrokinetic phenomena as related to the localized variation of the zeta potential and nonlinearity. We anticipate that our work would revitalize research related to the fabrication of alternate electrical voltage/power sources from liquid flow over charged surfaces. Other anticipated applications extend to electrophoretic applications in biological separations (cell transport, manipulation[38], and interactions[39]), as well as voltage sources for lab-on-a-chip applications[40].

In summary, our work has experimentally demonstrated the largest figure of merit thus far[41–44], to the best of our knowledge, with primary focus on methodologies related to enhance the streaming potential ($V_s$) per unit pressure difference ($\Delta P$) through the use of LFS (also see Supplementary Note 6 for considerations related to overall electrokinetic efficiency). The use of the LFS yields a figure of merit increase by a factor of 1.4 in comparison to that obtained using the AFS. It has been shown that larger voltages, through a measured streaming potential, may be achieved through careful engineering of the coupled electric field and fluid flow. The application of the related increase in the electrokinetic energy conversion efficiency would need further optimization of the fluidic and electrical impedances, in concert with the streaming conductance, as matched to an appropriate load[45]. Concomitantly, unipolar transport (where for example, either $Na^+$ or $Cl^-$ ions are transported[46]) through EDL overlap[26,47] in nanoscale channels may be coupled with LFS to yield much larger voltages, comparable to that of batteries[48].

## Methods

**Design considerations and fabrication of air-filled surfaces and liquid-filled surfaces.** The critical parameters in our channel surface design are (a) structural, i.e., the interstitial fraction of AFS/LFS: $\phi$-the ratio $[= d/(d+w)]$, with an average ridge width ($w$) and spacing ($d$), as indicated in Fig. 2a, (b) dimension, the channel length ($L$) and channel width ($W$) should be much larger than channel height ($H$), then the channel can be treated as two infinite parallel plates and the flow is Poiseuille flow, (c) pressure gradients, i.e., over a range of 0–1000 Pa, with higher pressures yielding greater streaming potentials ($V_s$), and (d) the nature of the filling fluid, easier for oil with low surface tension to penetrate into the grooves. For instance, we have used in our electrokinetic flow experiments, $\phi \sim 0.5$, 0.9 cm wide, 11.8 cm long, and 250 μm high channel, with a pressure difference of $\sim 1200$ Pa, using GPL oil as a filling liquid, to generate maximal $V_s$.

The LFS were fabricated through filling oil into the grooves of a patterned surface, fabricated through photolithography, and dry etching. A Si wafer (n-type, $< 110 >$, thickness 500 μm) was cleaned with acetone and IPA, rinsed by DI water, and subsequently baked at 180 °C for 5 min. Negative photoresist (NR9-3000) was coated on the Si wafer (3000 rpm for 40 s) and baked at 120 °C for 60 s. The pattern was defined through a mask (using the EVG 620) and developed (RD6) for 1 min. Then the resist-patterned wafer was subject to dry etching (using Oxford Plasmalab 100 RIE/ICP) to yield a trench depth ($h$) $\sim 95$ μm. The photoresist was then removed (using RR2 photoresist remover) over $\sim 12$ h and any residues were further removed (using PVA TePla PS100 at 120 sccm, 200 W for 90 s). The grooved surface was coated with Parylene-C (using PDS 2010 Parylene Coater) to a thickness of $\sim 600$ nm.

For LFS, oils, e.g., Dupont Krytox GPL 104, Castor oil, etc. was filled into the grooves of the patterned surface. The air-/liquid-patterned surfaces were imaged using environmental scanning electron microscopy (ESEM): FEI/Phillips

XL ESEM and the FEI Quanta FEG 250 ESEM. We have indicated, e.g., in Fig. 2c that a lower $\phi$-the ratio of the average groove width to the overall period yields a larger $V_s$.

**Choice of applied pressure and related flow rate.** The electrolyte flow rate in the channel, constituted from rectangular plate geometry[26], was modulated through a syringe pump, Fig. 1c, and was transduced to an applied pressure ($\Delta P$) based on the Poiseuille relation: $\Delta P = \frac{12\eta LQ}{wh^3}$ ($Q$ is flow rate, $L$ is channel length of ~11.8 cm, and $h$ the channel height of ~250 μm). The $\Delta P$ over the channel length (where the other end was left open to atmosphere) motivated the pressure gradient driven flow, and for the experiment was in the range of 200 to1200 Pa, as measured through a manometer (UEI EM152 Dual Differential Input Manometer). It was observed that when the $\Delta P$ was larger (/smaller) than 1200 Pa (/200 Pa), that the $V_s$ was not stable.

**Choice of liquid to fill the interstices in the liquid-filled surfaces.** For the fabrication of the LFS, the filling liquid (e.g., Krytox® GPL 104 and castor oil) was chosen primarily on the basis of immiscibility with water/electrolyte. Additional criteria would be a low surface tension—for better penetration into the interstices, low viscosity, and a smaller dielectric constant—to reduce electrolyte–oil interfacial shear stress. We observe (see Supplementary Figure 1) that the parylene-coated Si surface was oleophilic (/hydrophobic), and that the chosen liquid or oil could penetrate the interstices easily, Fig. 3a.

## Data availability

The data that support the plots within this paper and other findings of this study are available from the corresponding author upon reasonable request.

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

## Acknowledgements

The authors are grateful for support from the National Science Foundation (NSF: CMMI 1246800 and CBET 1606192). We also appreciate the assistance of Prof. J. Friend and P. Chen for help with imaging and Dr. S. Rubin for discussions.

## Author contributions

B.F. and P.R.B. conceived the ideas and experiments, which were mainly performed by B.F. A.B. contributed to the computational simulations. All the authors contributed to the analysis and interpretation of the results, and the writing of the paper.

## Additional information

**Competing interests:** The authors declare no competing interests.

