## [Peer Review File · Nature Communications]

Reviewers' comments:

Reviewer #1 (Remarks to the Author):

The work NCOMMS-18-08780, demonstrated a way to generate an enhance voltage via a patterned channel. This work is somehow novelty and interesting to the readers. However, in humble review of the referee, this work at current stage at an earlier stage and needs significant efforts to elaborate current results and further show more evidences and application demonstration to impact broader community rather than in microfluidic community

1, some theoretical background on Anisotropic electro-osmotic flow over super-hydrophobic surfaces has been done before in J. Fluid Mech. (2010), vol. 644, pp. 245–255.

2, what is the relation between theory and practical implementation in channel design and fabrication, which ones are critical parameters?

3, what the impact of the filled materials, size, material properties?

4, how to make stable and reliable filling of the liquid

5, which kind of knowledge of the work can impact to other fields rather than microfluidic community? proved applications or method, protocols?

Reviewer #2 (Remarks to the Author):

The manuscript describes an interesting approach to enhance the streaming potential by introducing periodic liquid field surfaces. The concept is based on the slip boundary condition, which is a widely investigated topic but mostly by theoretical approach. Therefore the manuscript is expected to draw wide attention in this particular field of research. However, the results and the discussion is presented in a way, which makes it hard to understand which of the proposed mechanisms/effects play the dominant role in the voltage enhancement. I, therefore, cannot recommend the publication the manuscript in its present form. However, if a resubmission is considered, the following major and minor issues as listed below must be adequately addressed .

Major Issues

1. What is the maximum energy conversion efficiency obtained with LFS.
2. What is the effect of the surface geometry alone on the observed streaming potential? Nano and micro-scale roughness is expected to significantly effect the electrokinatic flow. Has this aspect been considered?
3. Is the choice of the parameter used for the periodic grooves optimum? What is the relative effect of the height, width and the period of the groves on the generated streaming potential?
4. How the surface conductivity changes upon using periodic liquid filled surface?
5. Given that both GPL and Castor oil shows low contact angle with parylene surface, how accurate is the effective medium approach considered in S5? Should the oil not cover the parylene surface also at least partially?
6. Page 7, line 7; "... A significant enhancement of 50%, to around ~52 mV was observed..." In comparison to what?
7. Many parameters has been speculated to support the observed zeta potential such as the effect of the slip length, surface charge, van der Waals energy interaction, viscosity and dielectric constant, whereas very little evidence has been provided showing explicit dependence of the streaming potential on any of these parameters.
8. The claim of demonstrating the largest figure of merit should be supported by documentary evidence from the literature containing specific as well as reasonable similarity with the proposed technique.

Minor issues

1. The best figure of merit obtained with LFS, in comparison to AFS should be mentioned in the abstract and summary
2. Page 1, Line 9 (introduction); should be "...termed as streaming potential"
3. A better description of the patterned surface in Fig 1. Or in Fig.2 Or in the associated text should be given. For example, what is the substrate material, values of d,w and h as well as the coating material thickness.
4. Flow direction can be indicated in Fig.2 a.
5. Page 6, Line 6; Should be "... with an estimated b~10 μm "

Reviewer #3 (Remarks to the Author):

The authors present experiments and supporting numerical computations on the streaming potential generated by pressure-driven (Poiseuille) flow through a planar microchannel containing an electrolyte, where the bottom of the wall of the channel has striped grooves characteristic of a superhydrophobic surface. When the grooves are filled with air, the authors measure only a minor increase in the streaming potential relative to a flat (un-grooved) surface, which is

consistent with previous theory (Ref 4.). This is due to the fact that the air-electrolyte interface is uncharged and hence the increased slip there is offset by the reduction in charge. The novelty of the present work is to fill the grooves with a low dielectric constant liquid (oil) for which an appreciable increase in the streaming potential relative to a flat surface is measured for certain oils. The authors claim this occurs as the oil-electrolyte interface can support surface charge and slip. The paper ends with the claim that the present study demonstrates the largest figure of merit (voltage per pressure drop) observed to date for this system.

This is an interesting paper that I think is broadly suitable for publication in Nature Communications; I think it will be of interest to people in the fields of electrokinetics and microfluidics. I like that the authors have conducted experiments on this system, given that most previous work has been theoretical/computational. I would, however, ask the authors to consider the following comments:

1. Page1: The idea of converting mechanical to electrical energy via streaming potential was discussed in 1965 by Morrison and Osterle, J. Chem. Phys vol. 43, who cite a paper by Quincke from 1859! The authors should acknowledge that the phenomenon they are studying has a long history, by citing these papers (and perhaps others).

2. Bottom of page 2: taking 10mV as an estimate for zeta seems low; I suggest 25mv which is the thermal scale kT/e .

3. Page 4: I would not interpret a slip length (b) as leading to an increase in zeta potential, since the zeta potential is an equilibrium quantity. Rather, with slip the fluid velocity is increased such that the *apparent* zeta appears larger than the actual/equilibrium zeta by a factor of $1+b/\lambda_D$. The authors should also note that this $(1+b/\lambda_D)$ enhancement in fluid velocity breaks down in the presence of surface conduction due to e.g. surface curvature or surface inhomogeneity (such as the no-slip/slip transitions of their grooved surface), see Khair and Squires Phys. Fluids vol. 21 (2009) for a discussion of this.

4. Bottom of page 6 and also page 8: the authors assume that the liquid-filled section of the grooved surface has a Navier slip condition. But, isn't the correct condition that the jump in tangential stress across the interface is zero? (In the absence of Marangoni stresses.) The authors should justify why they can replace the continuity of stress condition with the simple assumption of a slip length in their computations.

5. Bottom of page 9: The computational model assumes the dielectric constant is reduced close to the surface (from 78 to 28). How is this implemented computationally? Is it a gradual change or a sharp variation? It is also known that the viscosity can change close to the surface due to the strong field in the double layer, just like the dielectric constant. Was this effect modeled? If not,

why is it justifiable to neglect the viscosity variation but still vary the dielectric constant?

6. Page 10: “Largest figure of merit” — it would be useful to discuss previous experimental work and the figure of merit predicted therein.

Responses to Reviewers:

Reviewer #1:

Comment: *The work NCOMMS-18-08780, demonstrated a way to generate an enhance voltage via a patterned channel. This work is somehow novelty and interesting to the readers.*

Response: We thank the reviewer for the encouragement and do hope that our work motivates scientists working in fluid mechanics as well as electrokinetics, as well as technologists looking for generating voltages *in situ* for efficient microfluidic devices.

Comment: *some theoretical background on Anisotropic electro-osmotic flow over super-hydrophobic surfaces has been done before in J. Fluid Mech. (2010), vol. 644, pp. 245–255.*

Response: Generally, there has been much less experimental work compared to analytical and computational research in the area of electrokinetic flows and our research aims to contribute to such avenues.

Consequently, while the primary focus in our paper has been experimental, we have attempted to provide adequate theoretical background with respect to the work by Squires (*e.g.*, *Physics of Fluids*, vol. 20, 092105, (2008), *J. Fluid Mechanics*, vol. 615, 323, (2008)), Zhao (*e.g.*, *Physics of Fluids*, vol. 23, 022003, (2011)) as also indicated by our references (*e.g.*, References 4, 14, 17, 2128, 29, 30, 31,34, and 36) in the main text. The suggested reference - S.S. Bahga, *et al*,

vol. 644, p. 244, (2010), mainly deals with gas trapped superhydrophobic surfaces and considers electro-osmotic mobility as a tensorial characteristic and has been cited in our manuscript.

Additional references have been added to the revised text of the manuscript to provide more theoretical background.

Comment: *what is the relation between theory and practical implementation in channel design and fabrication, which ones are critical parameters?*

Response: The critical parameters in our channel design are (a) *structural, i.e.*, the interstitial fraction of AFS/LFS: ϕ - the ratio [= $d/(d+w)$], with an average ridge width (w) and spacing (d), as indicated in Fig. 2(a), (b) *dimension*, the channel length (L) and channel width (W) should be much larger than channel height (H), then the channel can be treated as two infinite parallel plates and the flow is Poiseuille flow, (c) *pressure gradients, i.e.*, over a range of 0-1000 Pa, with higher pressures yielding greater streaming potentials (V_s), and (d) *the nature of the filling fluid*, easier for oil with low surface tension to penetrate into the grooves. For instance, we have used in our electrokinetic flow experiments, $\phi \sim 0.5$, 0.9 cm wide, 11.8 cm long and 250 μm high channel, with a pressure difference of ~ 1200 Pa, using GPL oil as a filling liquid, to generate maximal V_s .

The text of the manuscript has been revised to clearly indicate the critical parameters in the design of the textured air-filled surfaces (AFS) and liquid-filled surfaces (LFS). A related discussion was added to section S1 of the Supplementary Information, and cited in the main revised manuscript.

Comment: *what the impact of the filled materials, size, material properties?*

Response: We consider for the *filled materials, i.e.*, air for an air filled surface: AFS, and oil for the liquid filled surface: LFS. The impact of filled oil *compared to* filled air is that the liquid-oil interface charge may be controlled, with consequent and significant contribution to the streaming potential (V_s). For filled air, as indicated in Figure 2 (c), a smaller filling fraction generates a larger V_s .

In the case of LFS, the (a) dielectric constant (ϵ_r), as well as (b) the viscosity (η) of the filled liquids may be important parameters affecting the generated streaming potentials. In our experiments, the grooves were filled by liquids of the following ϵ_r , such as GPL oil ($\epsilon_r \sim 2.1$, $\eta \sim$

340 cP), castor oil ($\epsilon_r \sim 4.7$, $\eta \sim 312$ cP). Generally, the oil surface would have a greater velocity (u_s) with a smaller ϵ_{oil} . Since the ϵ_r is smaller for the GPL oil at ~ 2.1 , a larger u_s is expected compared to that in castor oil, with an ϵ_r (~ 4.7). The shear plane, corresponding to bulk fluid motion is relatively closer to the effective charged surface in the GPL oil. The related magnitude of the zeta potential (ζ) is hence larger through the use of the GPL oil, and manifests through the larger V_s – as indicated through the results of Figure 3(c). It was also found that the obtained V_s was larger for the higher viscosity oils, considering a possible correspondence between the η and the ϵ_r - as indicated for example in Kumar, *et al*, J. Food Sci. & Tech, vol. 50, p. 549, (2013).

A related discussion was added to the main text as well as section S3 of the Supplementary Information.

Comment: *how to make stable and reliable filling of the liquid*

Response: The grooves were filled with liquids, *i.e.*, oils, with sufficiently low surface tension (γ), compared to that of the electrolyte (with $\gamma_{ele} \sim 72$ mJ/m²), *e.g.*, GPL oil ($\gamma_{GPL} \sim 18$ mJ/m²) and castor oil ($\gamma_{castor} \sim 39$ mJ/m²). The oil was spread evenly on the surface (parylene coated with $\gamma_{parylene} \sim 46$ mJ/m²), and after a long enough time, any excess oil was removed from the top of the LFS, prior to filling with electrolyte on top.

From a surface energy point of view, the LFS is seen to be more stable compared to the AFS. For instance, the surface energy of the AFS: $E_{AFS} \sim r \cdot \gamma_{parylene} \sim 290$ mJ/m² with r as the roughness factor (= 6.3 from the ratio of the total area to the projected area, $\sim (18+96+18+96) \cdot w/(18+18) \cdot w$) – using the definition of r from Kim, *et al*, *Scientific Reports*, srep 37813, (2016), is much larger compared to the surface energy of the LFS: $E_{LFS} \sim 0.5 \times \gamma_{parylene} + 0.5 \times \gamma_{oil} \sim 32$ mJ/m². Consequently, the oil filling was found to be reliable and is stable for the LFS.

A related discussion was added to section S3 of the Supplementary Information.

Comment: *which kind of knowledge of the work can impact to other fields rather than microfluidic community? proved applications or method, protocols?*

Response: Our experiments have deep scientific implications underlying *fluid flow* interfacing with both air and liquid as well as fluid shear at a surface, and *electrokinetic phenomena* as related

to the localized variation of the zeta potential and nonlinearity. We anticipate that our work would revitalize research related to the fabrication of alternate electrical voltage/power sources from liquid flow over charged surfaces. Other anticipated applications extend to electrophoretic applications in biological separations (cell transport, manipulation, and reaction), voltage sources for lab-on-a-chip applications, *etc.*

A few papers related to biological application, *e.g.*, Li, *et al*, “*Transport, Manipulation, and Reaction of Biological Cells On-Chip Using Electrokinetic Effects*”, vol. 69, p. 1564, (1997), Jacobson, *et al*, “*Microfluidic devices for electrokinetically driven parallel and serial mixing*”, Analytical Chemistry, vol. 71, p. 4455, (1999), Zhang, *et al*, “*A Streaming Potential/Current-Based Microfluidic Direct Current Generator for Self-Powered Nanosystems*”, Advanced Materials, vol. 27, p. 6482, (2015) have now been cited in the revised manuscript.

The possible impact to other fields of science and technology, in addition to microfluidics and electromechanical systems, has been clearly indicated in the revised text of the manuscript. Several references have been added, as discussed above.

Reviewer #2:

Comment: *The manuscript describes an interesting approach to enhance the streaming potential by introducing periodic liquid field surfaces. The concept is based on the slip boundary condition, which is a widely investigated topic but mostly by theoretical approach. Therefore the manuscript is expected to draw wide attention in this particular field of research.*

Response: We thank the reviewer much for the positive comments and the encouragement. We do hope that our work would “draw wide attention” and stimulate further work from the scientific and technological community, in the broad area of microfluidics and electrokinetic phenomena. We do agree that there has been much less experimental work compared to analytical and computational research in the area of electrokinetic flows and our research aims to contribute to such avenues.

Comment: *What is the maximum energy conversion efficiency obtained with LFS.*

Response: Our focus, in the presented work, was less on the energy conversion efficiency and more on methodologies to enhance the streaming potential (V_s) per unit pressure difference (ΔP), i.e., $\left(\frac{V_s}{\Delta P}\right)$. Indeed, our work has experimentally demonstrated the *largest figure of merit*, thus far, in terms of the voltage generated per unit applied pressure. *Please see detailed response, at the end of this section, comparing the figure of merit with those from literature.*

More specifically, the fluid flow to electrical conversion efficiency ($Eff.$) = P_{out}/P_{in} , where the output power: $P_{out} (= \frac{1}{4}V_s \cdot I_s)$, $I_s (= V_s \frac{A\sigma}{L})$ is the streaming current, with $A (= w \cdot h)$ as the cross-sectional area of the channel of width: w , and height: h , and $P_{in} = Q \cdot \Delta P$, Q is the flow rate ($= \frac{Gh^3}{12\eta}$), with G as a constant pressure gradient, and η the viscosity – general formulation adapted from Olthuis, *et al*, *Sensors and Actuators B*, vol. 111-112, p. 385, (2005).

Then, the $Eff. = \frac{3V_s^2\sigma\eta w}{\Delta P L G h^3} = 3 \left(\frac{V_s}{\Delta P}\right) \left(\frac{V_s}{L}\right) \left(\frac{\eta\sigma}{G}\right) \left(\frac{w}{h^2}\right)$. For a given electrolyte concentration (fixed σ) and flow velocity (a given u), *both* $\left(\frac{V_s}{\Delta P}\right)$ and $\left(\frac{w}{h^2}\right)$ are important. While the former, i.e., $\left(\frac{V_s}{\Delta P}\right)$ was considered in detail in our work, the geometrical factor: $\left(\frac{w}{h^2}\right)$ may be significantly enhanced through the use of nanoscale diameter channels (small h) with the overlap of double layers, as seen for example, in the work by van der Heyden, *et al*, *Nanoletters*, vol. 6, p. 2232, (2006). We used channels of the order of 250 μm in height over which most of the electric field is

zero and a finite field is obtained only close to the channel surfaces of the order of $0.1 \mu\text{m}$ (the Debye length). Consequently, the estimated energy conversion efficiencies in our work is quite small of the order $10^{-3}\%$, employing the computational methodology indicated in van der Heyden, *et al*, *Nanoletters*, vol. 6, p. 2232, (2006). More specifically, the efficiency values are $\sim 7.7 \cdot 10^{-4}\%$ (through the use of the oil in a *LFS*), $\sim 3.8 \cdot 10^{-4}\%$ (on an *AFS*), and $\sim 3.5 \cdot 10^{-4}\%$ (on a flat unpatterned substrate).

An important point to note is that the energy conversion efficiency may be improved by more than a factor of two through the use of LFS compared to the AFS and flat/unpatterned substrates. We may expect that the efficiency of nanochannels may be further enhanced by using *LFS*, and will be one of the focus areas of future work.

Moreover, we had indicated in the manuscript that the “application of the related increase in the electrokinetic energy conversion efficiency would need further optimization of the fluidic and electrical impedances, in concert with the streaming conductance, as matched to an appropriate load. Concomitantly, unipolar transport (where for example, *either* Na^+ or Cl^- ions are transported) through electrical double layer overlap may be coupled with *LFS* to yield much larger voltages, comparable to that of batteries”.

The efficiency related aspects have been further clarified in the revised manuscript, and a related new section (Section 8) added to the Supplementary Information.

Comment: *What is the effect of the surface geometry alone on the observed streaming potential? Nano and micro-scale roughness is expected to significantly effect the electrokinetic flow. Has this aspect been considered?*

Response: For the influence of surface geometry alone, we can compare the results of *Flat* (unpatterned surface), and the air-filled surfaces (*AFS*): *Air_{0.5}* and *Air_{0.75}*, as indicated in Figures 2(c) and (d). The air in the *AFS* can reduce the friction between the flowing liquid electrolyte and patterned surface which helps in the increase of the streaming potential (V_s). However, a relatively less charged liquid-air interface will have little contribution to streaming potential, *e.g.*, as indicated in the papers by Squires, *Physics of Fluids*, vol. 20, p. 092105, (2008) and Bahga, *et al*, *J. Fluid Mechanics*, vol. 644, p. 245, (2010). Consequently, the V_s of *Air_{0.5}* is only a little larger than compared to the *Flat* case and the V_s of *Air_{0.75}* is even smaller, presumably due to an increase of non-charged liquid-air areas.

The aspect of nano- and micro-scale roughness was considered and found to be of much less importance compared to the groove width. The nominal roughness of the parylene coated surface was found to be $R_a = 1.25 \text{ nm} \pm 0.19 \text{ nm}$ (as determined through Dektak 150 Surface Profilometer) and the obtained streaming potential was insensitive to variations around such values. Generally, a roughness of $< 6 \text{ nm}$ has minimal effects on the slip length and flow rate, *e.g.*, as indicated in the paper by Zhu and Granick, *Physical Review Letters*, vol. 88, p. 106102, (2002) and can be considered hydrodynamically smooth, *e.g.*, as indicated in the paper by Cottin-Bizonne, *et al*, *Physical Review Letters*, vol. 94, p. 056102, (2005).

The text has been revised to clearly indicate the influence of surface geometry as well as micro-/nano- scale roughness aspects. A related discussion was added to the main text as well as section S1 of the Supplementary Information.

Comment: *Is the choice of the parameter used for the periodic grooves optimum? What is the relative effect of the height, width and the period of the grooves on the generated streaming potential?*

Response: We have indicated, *e.g.*, in Figure 2(c) that a lower ϕ - the ratio of the average groove width to the overall period, yields a larger V_s . For an LFS with a given $\phi_{solid} = 0.5$, data from more detailed measurements seems to indicate that a groove width of $\sim 18 \mu\text{m}$ (period = $36\mu\text{m}$) is close to optimal for obtaining the largest V_s see Figure S1(a), while the V_s was larger for the reported groove/trench height of $95 \mu\text{m}$: see Figure S1(b). It was noted that heights $> 95 \mu\text{m}$ lead to fragile structures.

Figure S1 The obtained streaming potential (V_s) as a function of the (a) groove width (w) - for $8 \mu\text{m}$, $18 \mu\text{m}$, and $40 \mu\text{m}$, and (b) groove/trench height (h) - for $60 \mu\text{m}$ and $95 \mu\text{m}$. In our paper, we report the results through the use of $w = 18 \mu\text{m}$, and a $h = 95 \mu\text{m}$.

We have also noted that for a given pressure gradient and ϕ , that the V_s increases with the number of periods of grooves. Further work is needed to critically understand such results, as well as the precise geometrical conditions related to the optimization, and we hope that our experimental work will simulate further inquiry, from the community, into considering such aspects.

The text of the manuscript has been revised to clearly indicate the possibility of obtaining maximal values of the steaming potential through optimizing parameters in the design, such as the groove width. A related discussion, and the data indicated above, was added to section S1 of the Supplementary Information.

Comment: *How the surface conductivity changes upon using periodic liquid filled surface?*

Response: The relative contribution of the surface conductivity (κ_s) to the bulk liquid conductivity (κ_b) may be parameterized through a Dukhin number, which is directly related to a length scale: L_H - termed a healing length, *e.g.*, Khair and Squires, *J. Fluid Mech.*, vol. 615, p. 323, (2008), over which there is a perturbation of the electric field and an influence of the surface conductivity. We estimated an L_H of $\sim 0.1 \mu\text{m}$, and given the relatively large values of the groove width of $\sim 18 \mu\text{m}$, the influence of the surface conductivity on the obtained results would be relatively small.

To a very small extent, the surface conductivity (κ_s) may be considered to have decreased through the use of oil of a lower surface charge density (σ) compared to the parylene coated solid surface, *i.e.*, σ_{oil} of $\sim 1.8 \text{ mC/m}^2$ compared to $\sigma_{\text{parylene}} \sim 3.6 \text{ mC/m}^2$, while there may be a relative increase compared to the use of air pockets, in an AFS.

A related discussion has been added to the revised manuscript.

Comment: *Given that both GPL and Castor oil shows low contact angle with parylene surface, how accurate is the effective medium approach considered in S5? Should the oil not cover the parylene surface also at least partially?*

Response: While significant care was taken in removing the oil from the surface in the use of the LFS, there is a possibility, as pointed out by the reviewer, that the oil may partially cover the

parlylene surface. Consequently, the effective medium approach (EMA) aspect may indeed be considered an approximation.

However, with respect to the discussion in the *Supplementary Information*, where we use the EMA, we previously considered a weighted summation of the ζ_{PDMS} , $\zeta_{parlylene}$, and $\zeta_{interstice}$ as follows:

$$\zeta_{eff} = 0.5\zeta_{PDMS} + 0.5\zeta_{parlylene} \quad (1a)$$

$$\zeta_{eff} = 0.5\zeta_{PDMS} + \frac{b_{eff}}{\lambda_D} (\phi_{parly} \zeta_{parlylene} + \phi_{interstice} \zeta_{interstice}) \quad (1b)$$

Previously, we used $\phi_{parlylene} = 0.25$ and $\phi_{interstice} = 0.25$ (assuming that parlylene area was not covered by oil) and obtained an effective slip length (b_{eff}) of ~ 44.9 nm for $GPL_{0.5}$. However, even with $\phi_{parlylene} = 0.1$ and $\phi_{interstice} = 0.4$ (assuming that 80% of the LFS was covered by oil), the b_{eff} is ~ 37.5 nm, a change of $\sim 16\%$. Consequently, the use of the EMA seems to be a reasonable approximation.

The text of the Supplementary Information (Section S5), related to the manuscript, has been revised to indicate the utility as well as the limitations of the effective medium approximation.

Comment: Page 7, line 7; “.... A significant enhancement of 50%, to around ~ 52 mV was observed...” In comparison to what?

Response: The comparison is related to the “significant enhancement” of the V_s on the LFS, with respect to the streaming potential obtained on the AFS, at the same value of fluid filling fraction (ϕ) of 0.5, in the context of Figure 3(b).

This aspect has been clarified in the revised manuscript.

Comment: Many parameters has been speculated to support the observed zeta potential such as the effect of the slip length, surface charge, van der Waals energy interaction, viscosity and dielectric constant, whereas very little evidence has been provided showing explicit dependence of the streaming potential on any of these parameters.

Response: We have based the explanation of our experimental results, on the linearity of the streaming potential (V_s) as a function of the applied pressure (ΔP) through the Helmholtz-

Smoluchowski relation: $V_S = \frac{\epsilon\zeta}{\eta\kappa} \Delta P$. The slope of the obtained experimental curves would then

be directly connected to the dielectric constant (ϵ) of electrolyte, the zeta potential (ζ), the electrolyte viscosity (η), and the bulk electrolyte conductivity (κ). Such parameters are then intimately connected, and independent variation would be difficult to discern experimentally. For instance, *both* the ϵ_r and η are intimately connected, *i.e.*, a decrease in the former may not be readily deconvoluted from an increase in the latter experimentally. Consequently, further theoretical or computational input (which is outside the purview of the present experimental work) would be necessary.

However, a few aspects related to the dependencies may be gleaned through experimental observations. For example, the proportional dependence of the V_s on slip length (b) can be delineated through Figures 2 (d), 3(b), where a larger b -(through the use of the *AFS/LFS* compared to flat surfaces) could be correlated to a larger V_s . Generally, a larger slip (larger b) would be directly correlated to a reduced friction, which implies reduced van der Waals forces/interactions between the surfaces (*AFS/LFS*) with respect to the moving electrolyte.

The dependence of the V_s on surface charge can be found from the comparison between *Air_{0.5}* and *GPL_{0.5}* in Figure 3 (b). For *Air_{0.5}*, the electrolyte-air interface is expected to have a lower interface charge compared to the electrolyte-oil interface (as in *GPL_{0.5}*) Consequently, the measured V_s of *GPL_{0.5}* is larger than that of *Air_{0.5}*.

We specifically indicated, *e.g.*, through the results of Figure 3(c), the influences of the relative dielectric constant (ϵ_r), as well as the viscosity (η) of the filling liquids on the generated V_s . Comparing GPL oil ($\epsilon_r \sim 2.1$, $\eta \sim 340$ cP) with castor oil ($\epsilon_r \sim 4.7$, $\eta \sim 312$ cP) yields a larger V_s in the former case. Here, the oil surface would have a greater velocity (u_s) with a smaller ϵ_{oil} due to larger van der Waals adhesion force at water-oil interface for oil with larger ϵ_{oil} . It was also found that the obtained V_s was larger for the higher viscosity oils in further experiment, considering a possible correspondence between the η and the ϵ_r - as indicated for example in Kumar, *et al*, J. Food Sci. & Tech, vol. 50, p. 549, (2013).

Such aspects have been clarified in the revised manuscript.

Comment: *The claim of demonstrating the largest figure of merit should be supported by documentary evidence from the literature containing specific as well as reasonable similarity with the proposed technique.*

Response: In our work, we achieved a figure of merit, defined as voltage generated per applied pressure, of 0.043 mV/Pa, which is much larger than previously measured. We indicate experimental reports indicating the figure of merit, below.

van der Heyden, *et al*, (*Nanoletters*, vol. 7, p. 1022, 2007) measured a streaming potential of 0.1 mM KCl solution flowing through a rectangular fused silica nanochannel with a width of 50 μm , length of 0.45 cm, and a height of 490 nm under a pressure driven flow. The obtained figure of merit was $\sim 1\text{V}/1\text{bar}$ yielding a value of $\sim 0.01\text{ mV}/\text{Pa}$.

Xie, *et al*, (*Lab Chip*, vol. 11, p, 4006, 2011) used two-phase flows, comprising injected nitrogen gas bubbles into a liquid (KCl electrolyte) filled microfluidic channel (44 cm long and 150 μm wide) in an effort to reduce the internal conduction current and enhance the power that may be obtained from electrokinetic flows. While a figure of merit of 0.030 mV/Pa was obtained there are issues related to difficulty in generating uniform flows in the presence of gas bubbles.

Xie, *et al.*, (*Applied Physics Letters*, vol. 93, p. 163116, 2008) monitored the V_s through the flow of KCl electrolyte (in the concentration range of $\sim 0.01\text{ mM}$ to 5 mM) in single track-etched nanopores of polyethylene terephthalate (PET) with an applied pressure of 1 bar (10^5 Pa). The corresponding figure of merit was in the range of 0.0002 mV/Pa to 0.0012 mV/Pa.

Li, *et al*, (*Plant and Soil*, vol. 386, p, 237, 2015) probed the flow of 0.1 mM NaCl solution driven through capillary channels (mimicking plant roots). The measured streaming potential increased linearly with applied pressure and a figure of merit of $\sim 0.005\text{ mV}/\text{Pa}$ was obtained.

Cited References: van der Heyden, *et al*, “Power Generation by Pressure-Driven Transport of Ions in Nanofluidic Channels”, (*Nanoletters*, vol. 7, p. 1022, 2007)

Xie, *et al*, “Strong enhancement of streaming current power by application of two phase flow”, (*Lab Chip*, vol. 11, p. 4006, 2011)

Xie, *et al*, “Electric energy generation in single track-etched nanopores”, (*Applied Physics Letters*, vol. 93, p. 163116, 2008)

Li, *et al*, “Zeta potential at the root surfaces of rice characterized by streaming potential measurements”, (*Plant and Soil*, vol. 386, p, 237, 2015)

All the related references have now been incorporated into the revised manuscript and have been discussed and cited.

Comment: *The best figure of merit obtained with LFS, in comparison to AFS should be mentioned in the abstract and summary*

Response: We have mentioned *both* in the abstract as well as the summary, of our revised manuscript, that the figure of merit obtained with the *LFS* is a factor of 1.4 larger than that obtained through the use of the *AFS*.

Comment: *Page 1, Line 9 (introduction); should be “termed as streaming potential”*

Response: The phrase “termed a streaming potential” has been changed to “termed as streaming potential”

Comment: *A better description of the patterned surface in Fig 1. Or in Fig.2 Or in the associated text should be given. For example, what is the substrate material, values of d, w and h as well as the coating material thickness.*

Response: We have indicated, in the revised text, a better description of the patterned surface in terms of clearly delineating the substrate, the parylene coating, as well as the used values and the range describing the geometrical parameters related to the width, height and period of the grooves. These changes have been made both in the text as well as the figure captions.

Comment: *Flow direction can be indicated in Fig.2 a.*

Response: The flow direction has been indicated clearly in the revised Figure 2(a).

Comment: *Page 6, Line 6; Should be “.... with an estimated $b \sim 10 \mu\text{m}$ ”*

Response: The text has been changed in accord with the suggestion.

Reviewer #3:

Comment: *This is an interesting paper that I think is broadly suitable for publication in Nature Communications; I think it will be of interest to people in the fields of electrokinetics and microfluidics. I like that the authors have conducted experiments on this system, given that most previous work has been theoretical/computational.*

Response: We thank the reviewer much for the encouragement and are in strong accord with his comments related to our work enhancing the interest of “*people in the fields of electrokinetics and microfluidics*”. We also agree that there has been much less experimental work compared to analytical and computational research in the area of electrokinetic flows and our research aims to contribute to such avenues.

Comment: *Page1: The idea of converting mechanical to electrical energy via streaming potential was discussed in 1965 by Morrison and Osterle, J. Chem. Phys vol. 43, who cite a paper by Quincke from 1859! The authors should acknowledge that the phenomenon they are studying has a long history, by citing these papers (and perhaps others).*

Response: We do acknowledge and agree completely with the reviewer that the generation of a voltage (*i.e.*, streaming potential) through flow of electrolyte over a surface has a “*long history*”.

Such an aspect along with the suggested references have been added, in the Introductory sections of the revised manuscript.

Comment: *Bottom of page 2: taking 10mV as an estimate for zeta seems low; I suggest 25mv which is the thermal scale kT/e .*

Response: Considering a revised zeta potential (ζ) of ~ 25 mV, in accord with the suggestion, yields a value of the streaming potential (V_s) of the order of 18 mV. It is to be noted, that in our experiments, that the obtained values of the V_s are significantly larger, *i.e.*, of the order of 36 mV on air filled groove surfaces – as in Figure 2(c), and ~ 52 mV on liquid filled surfaces – as in Figure 3(b).

The text has been revised to accommodate the new value of the zeta potential.

Comment: *Page 4: I would not interpret a slip length (b) as leading to an increase in zeta potential, since the zeta potential is an equilibrium quantity. Rather, with slip the fluid velocity is increased such that the *apparent* zeta appears larger than the actual/equilibrium zeta by a factor of $1+b/\lambda_D$. The authors should also note that this $(1+b/\lambda_D)$ enhancement in fluid velocity breaks down in the presence of surface conduction due to *e.g.* surface curvature or surface inhomogeneity (such as the no-slip/slip transitions of their grooved surface), see Khair and Squires *Phys. Fluids* vol. 21 (2009) for a discussion of this.*

Response: We do agree that the increased slip velocity (u_s) which is proportional to $\frac{\epsilon\zeta}{\eta}$, yields only an increase in the apparent zeta potential (ζ). Indeed, we have found through our own computational simulations that the streaming potential (V_s) enhancement is not directly related, and sensitive, to an increased b . For instance, with various b , *i.e.*, 0 μm , 10 μm , and 18 μm , we obtained V_s values of 51.7 mV, 53.1 mV, and 53.4 mV, respectively.

We also thank the reviewer for the suggested reference, for conditions related to non-uniform surface conduction, which may explain the significantly lower enhancements seen in our work.

The text in the manuscript has been revised to remove any implied correlation between the b and the ζ and the V_s .

Comment: *Bottom of page 6 and also page 8: the authors assume that the liquid-filled section of the grooved surface has a Navier slip condition. But, isn't the correct condition that the jump in tangential stress across the interface is zero? (In the absence of Marangoni stresses.) The authors should justify why they can replace the continuity of stress condition with the simple assumption of a slip length in their computations.*

Response: The continuity of stress condition is indeed appropriate, and is compatible with the used Navier slip condition, for the electrolyte-oil interface. In the electrolyte-air case, considering the very small dynamic viscosity (η) of air (~ 0.02 mPa·s) compared to that of the 0.1 mM electrolyte solution (with η of ~ 90 mPa·s) yields a very small shear rate in the electrolyte and implies a close to perfect slip (very large slip length: b) condition. However, when the air is replaced with oil (with η of ~ 300 mPa·s), the b is smaller.

Moreover, we have conducted the simulations considering the continuity of tangential shear stress across the electrolyte-oil interface (whereby the jump in the tangential stress is zero) and obtain a V_s very close (within 1 %) to the results previously obtained through an assumption of a slip length.

The text has been revised to indicate the use of the Navier slip boundary condition and compatibility with the zero tangential stress condition.

Comment: *Bottom of page 9: The computational model assumes the dielectric constant is reduced close to the surface (from 78 to 28). How is this implemented computationally? Is it a gradual change or a sharp variation? It is also known that the viscosity can change close to the surface due to the strong field in the double layer, just like the dielectric constant. Was this effect modeled? If not, why is it justifiable to*

neglect the viscosity variation but still vary the dielectric constant?

Response: The dielectric constant (ϵ_r) had been modeled as a step function, with a value of 28 over a distance of $\sim 3\lambda_D$ (~ 100 nm) proximate to the *LFS* and an $\epsilon_r \sim 78$ beyond - corresponding to the bulk solution. The distance is related to the scale over which a finite electric field is present, and we use ϵ_r only as a parameter to indicate the influence of the electric fields on the streaming potentials. However, as pointed out by the reviewer, both the (a) dielectric constant (ϵ_r), as well as (b) the viscosity (η) of the electrolyte close to the surface may be important parameters affecting the generated streaming potentials. However, as our experimental results were fitted through a linear relationship between the V_s and the ΔP using the relation: $V_s = \frac{\epsilon\zeta}{\eta\kappa} \Delta P$, and we may only determine the compounded quantity $\frac{\epsilon\zeta}{\eta\kappa}$ from the slope, *both* the ϵ_r and η are intimately connected, *i.e.*, a decrease in the former may not be readily deconvoluted from an increase in the latter experimentally. Consequently, theoretical or computational input (which is outside the purview of the present experimental work) would be necessary to yield further guidance.

To directly address the point raised by the reviewer, there is indeed a possibility of a viscosity increase, *e.g.*, as has been indicated in Lyklema, *et al*, “On the interpretation of electrokinetic potentials”, *J. Colloid Science*, vol. 16, p. 501, (1961) and through computational work by Wu & Qiao, “Physical origins of apparently enhanced viscosity of interfacial fluids in electrokinetic transport”, *Physics of Fluids*, vol. 23, p. 072005, (2011). In the latter work, it was indicated that the variation of viscosity may be more important for hydrophilic, rather than hydrophobic surfaces, in contrast to our work.

We have also carried out simulations considering viscosity variation near the surface through which it was observed that the equivalent *increase* in the η is a factor of ~ 2.5 to obtain the experimentally observed streaming potentials, assuming that the ϵ_r remains at the bulk value of ~ 78 , *i.e.*, the η is increased to ~ 2.19 mPa·s in the length scale of ~ 100 nm, as previously assumed.

The text has been revised to understand issues related to the importance of the combined variation of the ϵ_r as well as the η of the filling liquids in the developed models.

Comment: Page 10: “Largest figure of merit” — it would be useful to discuss previous experimental work and the figure of merit predicted therein.

Response: In our work, we achieved a figure of merit, defined as voltage generated per applied pressure, of 0.043 mV/Pa, which is much larger than previously measured. We indicate experimental reports indicating the figure of merit, below.

Van der Heyden, *et al*, (*Nanoletters*, vol. 7, p. 1022, 2007) measured a streaming potential of 0.1 mM KCl solution flowing through a rectangular fused silica nanochannel with a width of 50 μm , length of 0.45 cm, and a height of 490 nm under a pressure driven flow. The obtained figure of merit was $\sim 1\text{V}/1\text{bar}$ yielding a value of $\sim 0.01\text{ mV}/\text{Pa}$.

Xie, *et al*, (*Lab Chip*, vol. 11, p, 4006, 2011) used two-phase flows, comprising injected nitrogen gas bubbles into a liquid (KCl electrolyte) filled microfluidic channel (44 cm long and 150 μm wide) in an effort to reduce the internal conduction current and enhance the power that may be obtained from electrokinetic flows. While a figure of merit of 0.030 mV/Pa was obtained there are issues related to difficulty in generating uniform flows in the presence of gas bubbles.

Xie, *et al.*, (*Applied Physics Letters*, vol. 93, p. 163116, 2008) monitored the V_s through the flow of KCl electrolyte (in the concentration range of $\sim 0.01\text{ mM}$ to 5 mM) in single track-etched nanopores of polyethylene terephthalate (PET) with an applied pressure of 1 bar (10^5 Pa). The corresponding figure of merit was in the range of 0.0002 mV/Pa to 0.0012 mV/Pa.

Li, *et al*, (*Plant and Soil*, vol. 386, p, 237, 2015) probed the flow of 0.1 mM NaCl solution driven through capillary channels (mimicking plant roots). The measured streaming potential increased linearly with applied pressure and a figure of merit of $\sim 0.005\text{ mV}/\text{Pa}$ was obtained.

Cited References:

van der Heyden, *et al*, “Power Generation by Pressure-Driven Transport of Ions in Nanofluidic Channels”, (*Nanoletters*, vol. 7, p. 1022, 2007)

Xie, *et al*, “Strong enhancement of streaming current power by application of two phase flow”, (*Lab Chip*, vol. 11, p. 4006, 2011)

Xie, *et al*, “Electric energy generation in single track-etched nanopores”, (*Applied Physics Letters*, vol. 93, p. 163116, 2008)

Li, *et al*, “Zeta potential at the root surfaces of rice characterized by streaming potential measurements”, (*Plant and Soil*, vol. 386, p, 237, 2015)

All the related references have now been incorporated into the revised manuscript.

In conclusion, we have addressed all of the reviewer comments through careful reasoning as well as additional experiments and simulations. We thank very much the reviewers for their encouragement of the novelty of the paper, as well as their time and critical comments/queries, in the answering of which we have been able to further strengthen the physical and experimental rationale.

Reviewers' Comments:

Reviewer #1 (Remarks to the Author):

it is ok for me

Reviewer #3 (Remarks to the Author):

The authors have satisfactorily addressed the comments of my original review. Therefore, I can recommend the paper for publication in its present form.

The point-by-point response follows:

Reviewer #1:

Comment: *it is ok for me*

Response: We thank the reviewer for the encouragement.

Reviewer #3:

Comment: *The authors have satisfactorily addressed the comments of my original review. Therefore, I can recommend the paper for publication in its present form.*

Response: We thank the reviewer for the recommendation.

As indicated earlier, we hope that our work would significantly encourage further theoretical, computational, as well as experimental efforts in the microfluidics and electro-mechanical systems communities paving the way for a better understanding of fluid-surface interactions as well as novel technological application related to new energy and power sources.